# Molecular molds for regularizing Kondo states at atom/metal interfaces

Xiangyang Li [1,2], Liang Zhu[1,2], Bin Li[1], Jingcheng Li[1], Pengfei Gao[1], Longqing Yang[1], Aidi Zhao [1], Yi Luo [1], Jianguo Hou[1], Xiao Zheng [1✉], Bing Wang [1✉] & Jinlong Yang [1✉]

Adsorption of magnetic transition metal atoms on a metal surface leads to the formation of Kondo states at the atom/metal interfaces. However, the significant influence of surrounding environment presents challenges for potential applications. In this work, we realize a novel strategy to regularize the Kondo states by moving a CoPc molecular mold on an Au(111) surface to capture the dispersed Co adatoms. The symmetric and ordered structures of the atom-mold complexes, as well as the strong $d_\pi$–$\pi$ bonding between the Co adatoms and conjugated isoindole units, result in highly robust and uniform Kondo states at the Co/Au (111) interfaces. Even more remarkably, the CoPc further enables a fine tuning of Kondo states through the molecular-mold-mediated superexchange interactions between Co adatoms separated by more than 12 Å. Being highly precise, efficient and reproducible, the proposed molecular mold strategy may open a new horizon for the construction and control of nano-sized quantum devices.

[1] Hefei National Laboratory for Physical Sciences at the Microscale, Synergetic Innovation Center of Quantum Information and Quantum Physics, University of Science and Technology of China, Hefei, Anhui 230026, China. [2]These authors contributed equally: Xiangyang Li, Liang Zhu. ✉email: xz58@ustc.edu.cn; bwang@ustc.edu.cn; jlyang@ustc.edu.cn

Magnetic atoms are important building blocks of nano-devices for applications in quantum information and quantum computation. For realizing on-demand design and fabrication of nano-devices, it is crucial to achieve precise manipulation and tuning of local quantum states in these magnetic nanostructures[1–7]. This thus requires the characteristic quantum states stay robust upon variation of surrounding environment, and respond regularly and consistently under external manipulation.

Adsorption of a transition metal (TM) atom on a metal surface could lead to the formation of Kondo states at the atom/metal interface under sufficiently low temperatures, which has been affirmed by the substantially enhanced electric conductance across the interface[8,9]. Physically, the Kondo states originate from the screening of the local spin of $d$ electrons on the TM atom by the spins of itinerant electrons in the metal substrate. Recently, the precise control and tuning of Kondo states have become a focus of experimental efforts[10–17]. However, it is generally observed that the Kondo features vary greatly with the local chemical environment surrounding the TM atom, such as the coordinating ligands, the chemical dopants, the metal substrate, etc. This makes it difficult to produce Kondo states at atom/metal interfaces with regular sizes and shapes.

It is well-known that Kondo states emerge at a Co atom adsorbed on the fcc or hcp domain of Au(111) surface, whereas a Co atom adsorbed on the domain wall does not exhibit Kondo signature[8]. This indicates that the interaction between the Co atom and the gold substrate depends sensitively on the atomistic structure of the surface. Moreover, the Kondo states centered at a Co adatom can be significantly influenced by another Co adatom in its vicinity, because of the direct[9] or indirect[14,18] interatomic interactions. For instance, the substrate-mediated Ruderman–Kittel–Kasuya–Yosida (RKKY) interaction between two Co adatoms can change from antiferromagnetic to ferromagnetic as the interatomic distance varies, leading to enhanced or suppressed Kondo screening[4,19].

In many experiments, a TM adatom is bound to organic ligands to form an organometallic compound[20–23], and its local electronic configuration is significantly influenced by the surrounding. However, the related Kondo states do not necessarily have regular shapes, because the ligand field and the local spin distribution are highly susceptible to variations in the surrounding environment. For instance, for an iron phthalocyanine (FePc) molecule adsorbed on the Au(111) surface, the $dI/dV$ spectra measured at the on-top adsorption site exhibit Kondo signatures that are distinctly different from that at the bridge site[20]. In addition, the presence of a nearby FePc molecule can lead to conspicuous broadening or splitting of the Kondo conductance peak, which is again caused by the substrate-mediated RKKY interaction between the two local spin moments[21].

The above experimental observations indicate that the local chemical environment has a profound influence on the Kondo states formed at atom/metal interfaces. However, first-principles-based theoretical study on the Kondo states has remained rather scarce. This poses a serious challenge for potential practical applications—is there a way to regularize the Kondo states, so that they could exhibit uniform features that are insensitive to the variation of environment? To solve this problem, in this paper we propose a novel strategy, with which we successfully realize the regularization of Kondo states centered at the Co adatoms on an Au(111) surface.

Our strategy is motivated by the experience that a regular shape can be attained by using a rigid mold. Following this idea, we propose to use a planar molecule with conjugated rings to capture the dispersed TM adatoms on the metal surface. Each conjugated ring also serves as a mold, which regularizes the local spin distribution as well as the resulting Kondo states centered at the captured TM adatom. Consequently, the influence of environment beyond the range of the planar molecule is substantially weakened.

In this work, we choose to use a metal phthalocyanine (MPc) molecule, which possesses four isoindole units. Each isoindole unit can capture one Co adatom through the strong bonding interaction between the conjugated $\pi$-orbitals on the isoindole and the $d$ orbitals on the Co adatom. More interestingly, if there is more than one isoindole unit hosting a Co adatom, long-range superexchange interactions may emerge between these spatially separated Co adatoms, which allows for fine-tuning of Kondo states in a controllable manner.

## Results

**Construction of $K_n$(CoPc) on Au(111) surface**. Figure 1 illustrates the experimental procedure using an MPc molecule to capture the Co atoms dispersed on the Au(111) surface. As shown in Fig. 1a, a CoPc molecule exhibits a clear four-lobe pattern in the scanning tunneling microscopy (STM) image, with each lobe representing an isoindole unit. Pushed by the atomically sharp STM tip, the CoPc molecule can be quite freely moved on the surface. When the CoPc is pushed towards and finally in contact with a Co adatom, one of its four lobes captures the Co adatom and thus forms an atom-molecule complex (referred to as $K_1$ hereafter). In $K_1$ the Co atom is located underneath the molecular plane, and hence it interacts directly with both the isoindole unit and the gold surface. The $K_1$ complex can also be moved around on the surface, while its structure remains stable (see Fig. 1b). This also indicates that the Co atom is bound more tightly to the isoindole unit than to the surface.

Manipulated by the STM tip (see Supplementary Movie 1), a CoPc can capture up to four Co adatoms. Figure 1c demonstrates the sequential formation of the $K_{2'}$, $K_3$, and $K_4$ complexes from $K_1$. Here, $K_n$ denotes the complex with $n$ Co adatoms captured by a CoPc. $K_2$ and $K_{2'}$ are isomers, in which the two Co adatoms locate at the ortho and para positions, respectively. During the process, the CoPc molecule plays the role of Pac-Man, while the Co adatoms are "devoured" like Pac-Dots. The atomistic structures of these complexes are sketched in Fig. 1d, and the presence of the Co adatoms are verified by the topographic images depicted in Fig. 1e. Figure 1f depicts the Kondo maps of these complexes (see Supplementary Figs. 1 and 2). Clearly, a Kondo cloud emerges at the outer region of each occupied isoindole unit, while the cobalt ion at the center is Kondo inactive. It is remarkable that all the Kondo clouds have a similar crescent shape, except in $K_{2'}$ where the two clouds appear more isotropic. This thus highlights the strong regularization effect of the CoPc molecular mold.

**Experimental measurement of $dI/dV$ spectra and $T_K$**. We now examine more closely the Kondo characteristics by investigating the measured differential conductance ($dI/dV$ versus $V$) spectra. As shown in Fig. 2a, the measured spectra of a bare Co adatom exhibit a Fano line shape depending sensitively on the adsorption domain. The Fano line shape originates from the quantum interference between two electron conduction channels[24]: the through-space and through-atom channels. In contrast, the measured spectra of a $K_1$ complex exhibit a single-peak structure independent of the adsorption domain. This is because the CoPc molecule blocks the through-space tunneling channel and thus quenches the Fano interference[24,25]. The Kondo temperature $T_K$ is extracted from the width of the measured Kondo conductance peak ($\Gamma$), followed by deconvolution of thermal and instrument-induced broadening (see Methods).

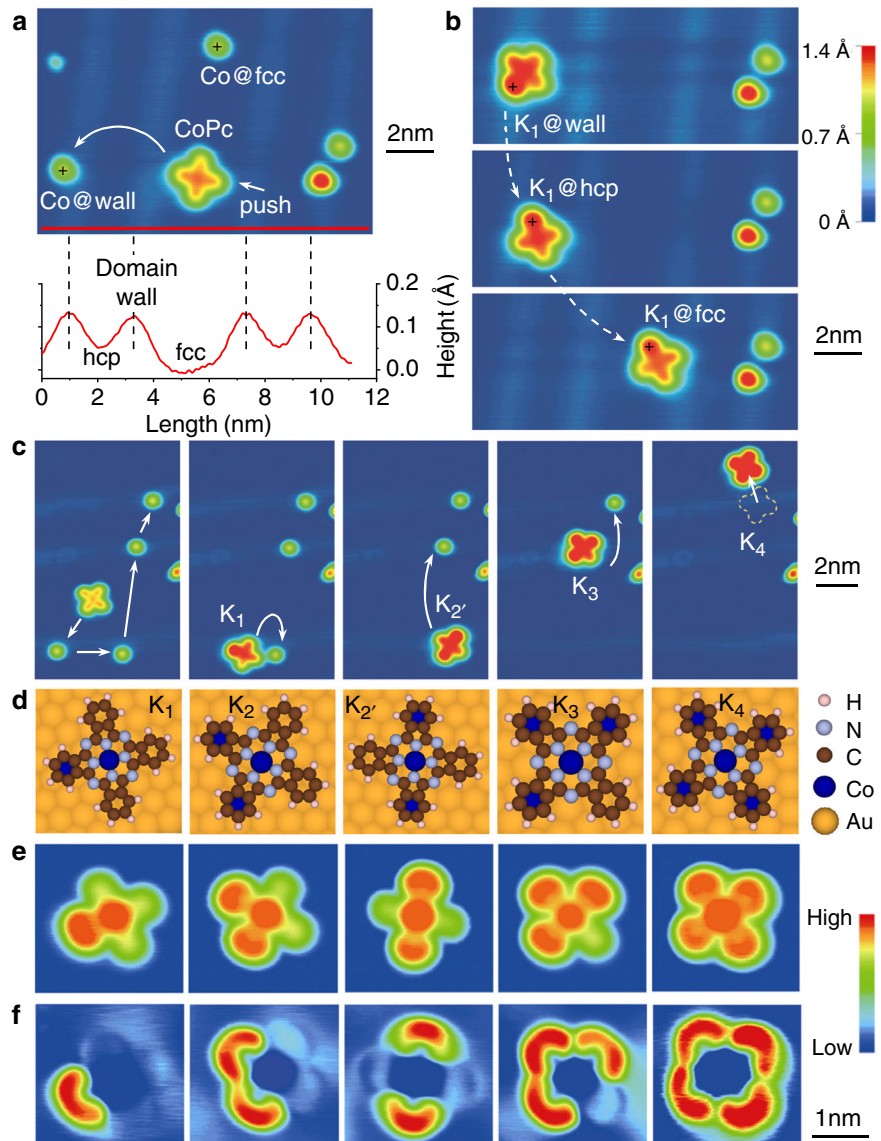

**Fig. 1 STM images for the formation of the $K_n$ complexes. a** Upper panel: a pristine CoPc and dispersed Co adatoms at different domains of the Au(111) surface. Lower panel: height profile of the Au(111) surface along the red line marked in upper panel. **b** Moving the $K_1$ complex across different domains of the Au(111) surface. **c** Consecutive formation of $K_n$ ($n = 1, 2', 3, 4$) complexes by precisely controlling the STM tip. **d** Structural models for the $K_n$ complexes, where each captured Co adatom is located underneath the six-member ring of an isoindole unit. **e** Topographic images and **f** Kondo maps of the $K_n$ complexes. See Supplementary Fig. 1 for more details.

From Fig. 2a, an isolated $K_1$ has an almost constant $T_K$ of about 122 K. By manipulating the STM tip, a $K_1$ can be moved towards a bare Co adatom. Throughout this process, the spectral line shape of the $K_1$ remains largely unchanged, while the $T_K$ exhibits a weak oscillation around the constant value (see Fig. 2b and Supplementary Fig. 3). Such an oscillation is supposed to originate from the substrate-mediated RKKY interaction between the Co adatom in the $K_1$ and the nearby bare Co adatom[4,19], and it is greatly suppressed with the nearby bare Co adatom captured by a CoPc mold. This affirms that the strong interaction between a Co adatom and the associated isoindole unit on the CoPc gives rise to highly regular and robust local spin distribution, so that the Kondo states are hardly affected by the variation of environment beyond the phthalocyanine ring.

For a $K_n$ complex hosting two or more Co adatoms, the $dI/dV$ spectra measured at each of the Co adatoms are almost identical (see Supplementary Fig. 2). While the spectral line shape of a $K_n$

($n \geq 2$) much resembles that of a $K_1$, the Kondo peak of the former is somewhat wider, suggesting a higher $T_K$ (see Fig. 2c). Figure 2d gives an overview of $T_K$ for all members of the $K_n$ family. Two main features are easily recognized: (1) $T_K$ increases linearly with $n$ for $n = 1, 2, 3, 4$. (2) $T_K$ of $K_{2'}$ is distinctly higher than that of its isomer $K_2$, and lies outside the above linear relationship. These features indicate that the Kondo state centered at a captured Co adatom is affected subtly but regularly by the other Co adatoms confined within the same phthalocyanine ring. This lays the foundation for fine-tuning the Kondo states formed at atom/metal interfaces through precise arrangement of the TM adatoms.

**Theoretical calculations of $dI/dV$ spectra and $T_K$.** The above experimental manipulations and measurements affirm the efficacy of CoPc molecular mold for regularizing the Kondo states at

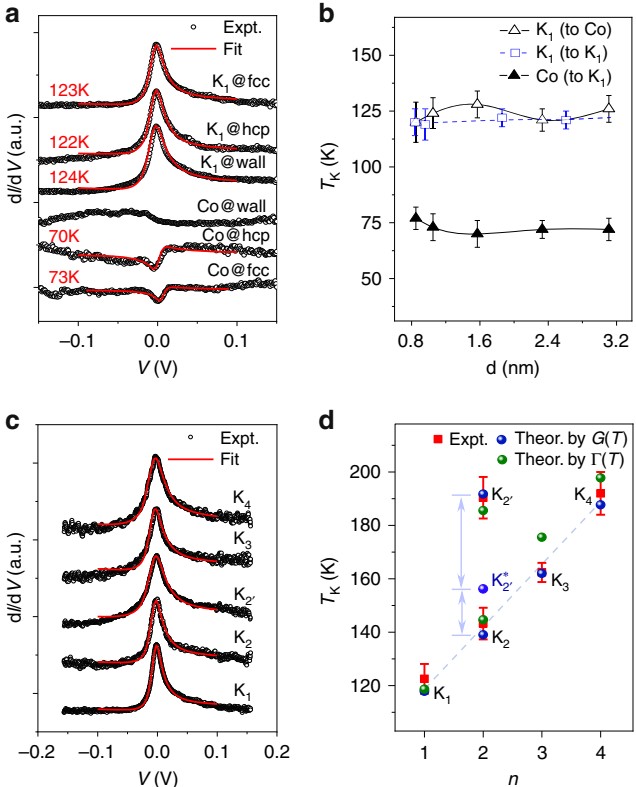

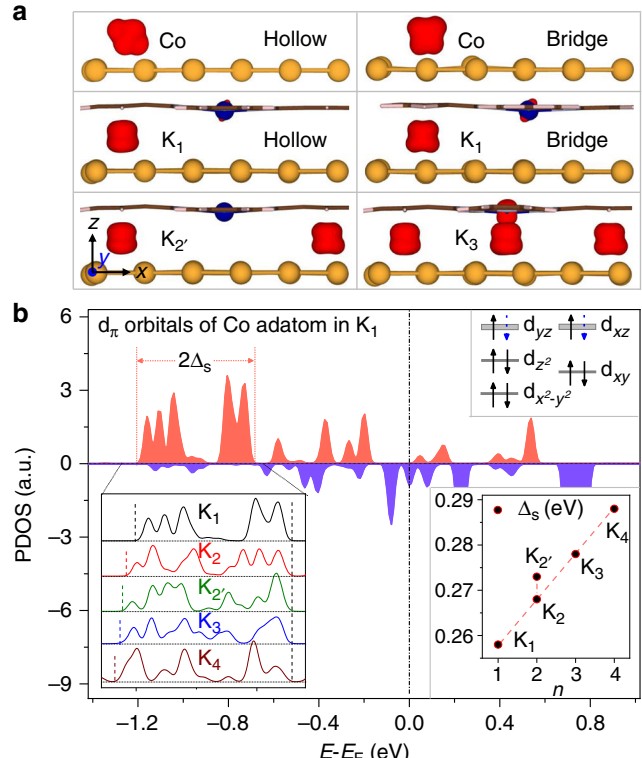

**Fig. 2 Kondo characteristics of the $K_n$ complexes. a** Measured $dI/dV$ spectra of a bare Co adatom and a $K_1$ adsorbed on different domains of the Au(111) surface. **b** Variation of $T_K$ of a $K_1$ (or a bare Co adatom) versus its distance to another $K_1$ (or a bare Co adatom) on the Au(111) surface. **c** Measured $dI/dV$ spectra of the $K_n$ complexes ($n = 1, 2, 2', 3, 4$). The red lines are fits of the measured data to the Fano function (see Methods for details), and they have been shifted vertically for clarity. The measured data are acquired by placing the STM tip at positions marked by the black crosses in Fig. 1a, b under the background temperature $T = 5$ K. **d** $T_K$ of the $K_n$ complexes extracted from experimentally measured and theoretically calculated $dI/dV$ spectra. The red bars mark the standard deviations of experimental data. The dashed line is a linear fit of the measured $T_K$ of $K_n$ ($n = 1, 2, 3, 4$). The circles (triangles) represent the values of $T_K$ determined by the heights (widths) of the calculated Kondo conductance peaks.

**Fig. 3 Local spin states of the Co adatoms in $K_n$/Au(111). a** Spin density distribution of a bare Co atom and a $K_n$ ($n = 1, 2', 3$) adsorbed on different sites of the Au(111) surface. The isosurface of 0.05 Å$^{-3}$ is shaded in red. **b** PDOS of $d_\pi$ orbitals of the captured Co adatom in a $K_1$. Positive (negative) values represent spin-up (spin-down) electrons. $\Delta_s$ is the half-width of the split peaks which constitute the main contribution to the local spin moment. The upper right inset is an orbital diagram showing the relative position and electron occupancy of every $d$ orbital of the Co adatom, where the thickness of a line indicates the broadening of the orbital by its surrounding environment, and a solid (dotted) arrow represents one (half an) electron residing on the orbital. The lower left inset displays the PDOS of $d_\pi$ orbitals of a captured Co adatom in each of the $K_n$ complexes ($n = 1, 2, 2', 3, 4$). The dashed lines mark the full width of the main peaks, which are shifted and aligned horizontally for a clear comparison. The lower right inset depicts the variation of $\Delta_s$ versus $n$.

Co/Au(111) interfaces. For practical purposes, it is crucial to elucidate the regularization mechanisms, e.g., what is the origin of the regular spectral line shapes, and what leads to the intriguing $T_K$ versus $n$ relationship. To provide theoretical insights into these questions, we carry out first-principles-based calculations by a combination of density functional theory (DFT)[26,27] and hierarchical equations of motion (HEOM)[28–30] approach. In Fig. 2d, the $T_K$ of $K_n$ extracted from the calculated $dI/dV$ spectra are compared against the experimental data (to be elaborated later).

Figure 3a depicts the optimized geometries and spin density distribution of a bare Co atom and a $K_n$ complex ($n = 1, 2', 3$) adsorbed on the Au(111) surface. Apparently, the planar CoPc mold is always parallel to the surface, and the captured Co adatom is right beneath the six-member ring of an isoindole unit. Such a geometry remains unchanged upon moving the $K_n$ to different sites on the surface (Supplementary Figs. 4–6). The spin-unpaired electrons reside predominantly on the Co adatoms with only a small fraction on the central cobalt ion. Thus, each Co adatom becomes the center of a Kondo cloud, whereas the cobalt ion on the CoPc mold is Kondo inactive.

From Fig. 3a it is also noted that the local spin density of a bare Co adatom exhibits an appreciable change at different adsorption sites. In contrast, the local spin of a captured Co adatom in a $K_n$ is hardly affected by the surrounding environment (see also Supplementary Figs. 7 and 8). These results are consistent with the experimental findings as displayed in Fig. 2a, b. Therefore, the CoPc mold indeed regularizes and preserves the local spins of the Co adatoms, and thus facilitates the formation of uniform Kondo states at the $K_n$/Au(111) interfaces.

By analyzing the projected density of states (PDOS) of captured Co adatoms, we obtain a schematic diagram showing the relative energy, broadening and electron occupancy of each $d$ orbital; see the upper right inset of Fig. 3b. The local electron configuration of a captured Co adatom is determined to be $d_{xy}^{1.6} d_{x^2-y^2}^{1.6} d_{z^2}^{1.8} d_{xz}^{1.3} d_{yz}^{1.2}$ (see Supplementary Table 1), and the local spin moment originates mainly from the $d_\pi$ ($d_{xz}$ and $d_{yz}$) orbitals. Thus, a captured Co adatom is considered to be in a local $S = \frac{1}{2}$ state, in clear contrast to the local $S = 1$ state of a bare Co adatom[31].

Because of their common symmetry, the $d_\pi$ orbitals on a Co adatom and the conjugated $\pi$-orbitals on an isoindole unit of the

CoPc mold interact strongly with each other. The resulting $d_\pi$–$\pi$ bonds further consolidate the atom-mold complex, and give rise to the dumbbell-shaped spin density distribution (see Fig. 3a). The physical origin of the regularization effect is thus clear. Confined by the CoPc mold, the local electronic structure of a Co adatom is significantly reshaped and regularized by the strong $d_\pi$–$\pi$ bonds, leading to a local spin state distinctly different from that of a bare Co adatom.

The strength of Kondo correlation depends critically on how strongly the local spin on a Co adatom is screened by the electronic spins in the surrounding environment. For the $K_n$/Au (111) composites, the Kondo screening is realized via the hybridization of the Co $d_\pi$ orbitals with the $s$ orbitals of the nearby gold atoms. It is such $s$–$d$ hybridization that leads to the broadening and splitting of the Co $d_\pi$ orbitals as shown in Fig. 3b and Supplementary Fig. 9. The hybridization strength is characterized by $\Delta_s$, which is taken as the half-width of the split PDOS peaks that constitute the main contribution to the local spin moment.

With more Co adatoms captured by a same CoPc, the split peaks of PDOS are slightly more broadened. This signifies a perturbation by the other Co adatoms confined within the same CoPc mold, possibly through the substrate-mediated RKKY interactions. The magnitude of such a perturbation is thus roughly proportional to the number of Co adatoms inside the atom-mold complex. Intriguingly, the split peaks of a $K_{2'}$ is somewhat broader than a $K_2$, implying the symmetry of the complex has a subtle influence on the strength of RKKY interaction. Accordingly, $\Delta_s$ increases linearly with $n$ except for $n = 2'$ (see the lower right inset of Fig. 3b). The apparent resemblance between the $\Delta_s$ versus $n$ and $T_K$ versus $n$ relationships (cf. Fig. 2d) affirms the key importance of $\Delta_s$ to the characteristic features of the Kondo states[32].

To quantitatively describe the Kondo effect, we employ quantum impurity models which explicitly include the electron–electron interactions. We first assume the Kondo clouds centered at different Co adatoms are independent of each other, and thus a tip/$K_n$/Au(111) junction can be represented by a single-orbital Anderson impurity model (AIM)[33]. The calculated $dI/dV$ spectra agree remarkably with the experimental measurements (see Supplementary Fig. 10). The $T_K$ of $K_n$ are determined based on the height or the width of the Kondo conductance peaks (see Methods and also Supplementary Figs. 11 and 12), and the theoretical values accurately and consistently reproduce the experimental data (see Fig. 2d).

With the single-orbital AIM, the $T_K$ of a $K_{2'}$ is predicted to be 157 K (labeled by $K_{2'}^*$ in Fig. 2d), only slightly higher than that of a $K_2$, yet considerably lower than the experimental value of 191 K (Supplementary Fig. 13 and Supplementary Table 3). This indicates that the use of a single-orbital AIM is inadequate, because the two Kondo clouds at the $K_{2'}$/Au(111) interface are not independent to each other.

### Influence of long-range superexchange interaction on $T_K$.
To gain deeper insights into the unusually high $T_K$ of a $K_{2'}$, we perform additional experiments by replacing CoPc with CuPc and $H_2$Pc molecules (see Supplementary Note 12 for details). Similar atom-mold complexes are constructed by manipulating the STM tip. Unlike the case of CoPc, the measured $T_K$ of $K_{2'}$ (CuPc) and $K_{2'}$($H_2$Pc) are only slightly higher than their isomers $K_2$(CuPc) and $K_2$($H_2$Pc) by 4–5 K (see Supplementary Figs. 14 and 15). These findings highlight the unique role of the central cobalt ion in the CoPc, which may serve as a hub to assist the two Co adatoms in the para positions to interact with each other.

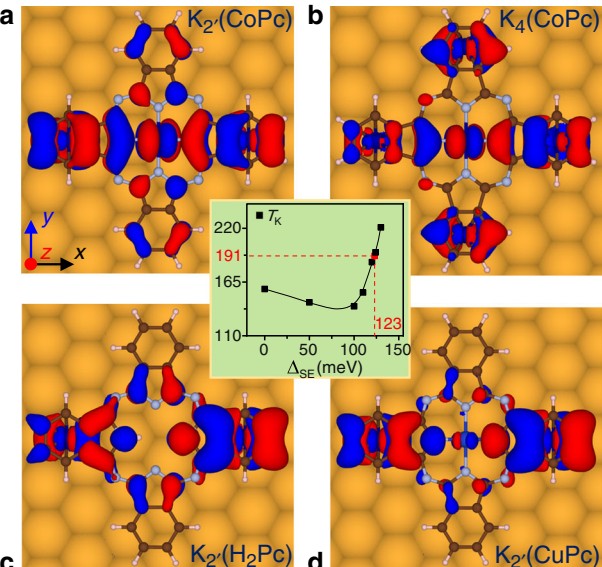

**Fig. 4 Delocalized Kohn–Sham orbitals in $K_n$(TMPc). a** A delocalized orbital at $E = -0.37$ eV in a $K_{2'}$(CoPc), **b** at $E = -0.44$ eV in a $K_4$(CoPc), **c** at $E = -0.66$ eV in a $K_{2'}$($H_2$Pc), and **d** at $E = -0.50$ eV in a $K_{2'}$(CuPc), respectively. The Fermi energy $E_F$ is set to zero. The isosurfaces of $\pm 0.001$ Å$^{-3}$ are shaded in red and blue, respectively. The inset shows how the $T_K$ of a $K_{2'}$(CoPc) varies with the superexchange interaction strength between the two captured Co adatoms ($\Delta_{SE}$), where the red square marks the predicted value of $\Delta_{SE} = 123$ meV for a $K_{2'}$(CoPc).

From the calculated electronic structures of $K_{2'}$(CoPc)/Au(111) and $K_4$(CoPc)/Au(111) composites (see Supplementary Figs. 16 and 17), a delocalized Kohn–Sham orbital, which links the $d_\pi$ orbitals on the two distantly separated Co adatoms via the $\pi$-orbitals on the isoindole units and a $d_\pi$ orbital on the central cobalt ion, is recognized (see Fig. 4a, b). This thus provides a channel for the superexchange (SE) interaction[34,35] between the two separated local spins. However, such a channel breaks down in a $K_{2'}$($H_2$Pc) or a $K_{2'}$(CuPc) on the Au(111) surface, because the bridging orbital at the central hub is absent or energetically incompatible (see Fig. 4c, d). Consequently, the long-range SE interaction is strong in a $K_{2'}$(CoPc), but either absent or rather weak in a $K_{2'}$($H_2$Pc) or $K_{2'}$(CuPc).

To assess the influence of SE on the Kondo states in a $K_{2'}$, a two-orbital AIM is employed, which explicitly includes the molecular-mold-mediated SE interaction between the two Co adatoms. Accurate computation of the strength of SE interaction ($\Delta_{SE}$) by ab initio quantum chemistry methods is highly desirable yet rather challenging. Instead, $\Delta_{SE}$ is determined by exploring the variation of $T_K$. To retrieve the experimental value of $T_K = 191$ K (see the inset of Fig. 4), we have $\Delta_{SE} = 123$ meV for a $K_{2'}$(CoPc), which amounts to an effective ferromagnetic coupling[36] of $J_{eff} \approx 4$ meV between the two local spins separated by as far as 12.5 Å (see Supplementary Notes 6 and 15 for details).

As shown in Fig. 4, the $T_K$ of a $K_{2'}$(CoPc) exhibits a nonmonotonic dependence on $\Delta_{SE}$. Such a phenomenon can be understood as follows. As $\Delta_{SE}$ increases from zero, the local spin on a Co adatom starts to feel the ferromagnetic interaction from the other Co adatom, and thus it is screened less strongly by the surrounding environment, which leads to the attenuated Kondo correlation. However, when $\Delta_{SE}$ reaches a certain value (see Supplementary Fig. 18), the whole $K_{2'}$(CoPc) complex favors an $S = 1$ state, resulting in a substantially enlarged spin moment for the total complex. Consequently, the Kondo temperature rises again with further increasing of $\Delta_{SE}$.

Among all the other $K_n$, only $K_4$(CoPc) possesses a delocalized SE channel. This indicates that the structural symmetry of the whole complex is vital for the existence of SE. Nevertheless, the SE in a $K_4$ is much weaker than in a $K_{2'}$ (see Supplementary Fig. 19), because the delocalized orbital involves also the two disconnected Co adatoms at the ortho positions. It is estimated that the SE could lead to a minor change in $T_K$ by $\sim 4$ K for a $K_4$.

Besides the strength of Kondo correlation, the SE also affects the spatial distribution of Kondo states. As noted earlier, the Kondo clouds at the $K_{2'}$/Au(111) interface appear to be more isotropic than in the other composites, which is possibly due to the strong long-range SE.

## Discussion

To summarize, unlike all the previous studies in which the Kondo states formed around an adsorbed TM atom always vary sensitively to the surrounding environment, we demonstrate that, the proposed strategy of using a symmetric and conjugated molecular mold gives rise to highly robust and regularized Kondo states. Adjusting the chemical composition of the molecular mold further allows for a fine-tuning of the Kondo characteristics. The resulting atom-mold complexes can move freely on the gold surface and exhibit remarkably uniform Kondo features. Therefore, they may serve as standard building blocks for the design and fabrication of novel quantum devices.

## Methods

**Scanning tunneling microscopy experiments.** The experiments are carried out with a low temperature STM (Omicron) using Au(111)/mica substrates in vacuum under a base pressure of $3 \times 10^{-11}$ Torr. The Au(111) surface is cleaned by repeat cycles of Ar ion sputtering at 800 V for 15 min and annealed at 600 K. Submonolayers of Co atoms and CoPc molecules are co-evaporate by e-beam evaporation and by sublimation, respectively, on an Au(111) substrate. The substrates are placed in situ on the cryostat of the microscope with a temperature of about 6.7 K. The samples are then investigated in situ at 5 K. The chemically etched tungsten tip is carefully cleaned by circular Argon ion sputtering at 800–1000 V for about 5–10 min and by field emission lasting for about 30 s under a sample bias of around $-100$ V with emission current of 2 μA in each cycle[37,38].

In the consecutive formation of $K_n$ complexes, the manipulation conditions are 50 mV and 10 nA, while the imaging conditions are 1 V and 0.2 nA for Fig. 1a–c. For the measurement of topographic images and simultaneously acquired Kondo maps in Fig. 1e–f, the imaging conditions are $-15$ mV and 2 nA with a sinusoidal modulation of 2 mV and 730 Hz. The $dI/dV$ spectra in Fig. 2 are recorded with a lock-in method. The acquisition conditions are $-150$ mV and 2 nA with a sinusoidal modulation of 2 mV and 730 Hz. The Kondo maps and $dI/dV$ spectra of the $K_n$ complexes are acquired at different domains of the Au(111) reconstruction surface, including hcp and fcc domains and the domain walls.

The Kondo line shapes in the $dI/dV$ spectra are fit to a single Fano function[39] as follows:

$$\frac{dI}{dV} \propto \frac{(q + \varepsilon)^2}{1 + \varepsilon^2}, \tag{1}$$

where $\varepsilon = (eV - \varepsilon_K)/\Gamma$, $\varepsilon_K$ is the center of the Kondo resonance, $\Gamma$ is the half-width at half-maximum (HWHM) of the Kondo resonance peak, and $q$ is the asymmetry parameter. The Kondo temperature $T_K$ is extracted from $\Gamma$ by deconvolution of the thermal and instrument-induced broadening as follows[40–44]:

$$k_B T_K \approx \sqrt{\Gamma^2 - (\lambda k_B T)^2 - (0.87 e V_m)^2}. \tag{2}$$

Here, $k_B$ is the Boltzmann's constant, $T = 5$ K is the environmental temperature, the coefficient $\lambda$ assumes an empirical value of 2.7 as adopted in previous experiments[42], and $V_m = 1$ mV is the amplitude of the modulation voltage (corresponding to a peak-to-peak value of 2 mV).

It is worth pointing out that in principle the thermal and instrument-induced broadening of any spectroscopic feature should be examined by a convolution of the expected signal with a proper broadening kernel[40–43]. In practice it is often more convenient to use the approximate formula of Eq. (2).

**Density functional theory method.** First-principles calculations are performed by using the spin-polarized density functional theory (DFT) method implemented in the Vienna ab initio simulation package (VASP)[45]. The generalized gradient approximation developed by Perdew, Burke and Ernzerhof (PBE)[46] is chosen for the exchange-correlation functional. The projected augmented wave method is

adopted with the energy cutoff of 400 eV. The zero damping DFT-D3 method of Grimme[47] is used to improve the description of van der Waals interactions.

During the structural optimizations, a slab model including three layers of Au atoms is used to represent the Au(111) surface, with each layer containing 56 Au atoms, A vacuum space of 16 Å is used to eliminate the interactions between different slabs. All the atoms except those in the bottom two Au layers are fully relaxed until the residual force on every atom was <0.02 eV Å$^{-1}$. The convergence criterion is set to $1 \times 10^{-5}$ eV for the total energy. Because of the large size of the supercell, the $\Gamma$-point approximation is adopted. Additional numerical tests have been carried out, which have verified that a larger slab model consisting of four atomic layers does not change the main results qualitatively.

**Anderson impurity models.** The AIMs[33] which explicitly include the electron–electron Coulomb interaction and the Kondo effects are adopted to represent the tip/$K_n$/Au(111) junctions. The total AIM Hamiltonian is comprised of three parts:

$$H_{AIM} = H_{env} + H_{imp} + H_{coup}, \tag{3}$$

where $H_{imp}$ and $H_{env}$ are the Hamiltonian of the magnetic impurity and that of its environment, respectively.

In particular, the environment is modeled by reservoirs of noninteracting electrons, i.e., $H_{env} = H_{tip} + H_{sub} = \sum_{\alpha = t,s} \sum_\sigma \epsilon_{\alpha k} \hat{c}^\dagger_{\alpha k \sigma} \hat{c}_{\alpha k \sigma}$, where $H_{tip}$ and $H_{sub}$ represent the STM tip ($\alpha = t$) and the gold substrate ($\alpha = s$), respectively. $\hat{c}^\dagger_{\alpha k \sigma}$ ($\hat{c}_{\alpha k \sigma}$) is the creation (annihilation) operator for a spin-$\sigma$ electron on the $k$th orbital in the $\alpha$th reservoir. The impurity-environment coupling has the form of $H_{coup} = \sum_{\alpha \sigma k i} t_{\alpha k i} \hat{c}^\dagger_{\alpha k i} \hat{d}_{i\sigma} + \text{H.c.}$, where $\{t_{\alpha k i}\}$ are the coupling strengths between the $i$th impurity orbital and the $k$th reservoir orbital. The influence of electron reservoirs on the impurity is accounted for through the hybridization functions defined by $\Lambda_{ij\alpha}(\omega) \equiv \pi \sum_k t_{\alpha k i} t^*_{\alpha k j} \delta(\omega - \epsilon_{\alpha k})$.

In a single-orbital AIM, the spin-polarized $d_\pi$ orbitals on each captured Co adatom are represented by a single impurity orbital, which is described by

$$H_{imp} = \epsilon_d (\hat{n}_\uparrow + \hat{n}_\downarrow) + U \hat{n}_\uparrow \hat{n}_\downarrow. \tag{4}$$

Here, $\hat{n}_\sigma = \hat{d}^\dagger_\sigma \hat{d}_\sigma$, where $\hat{d}^\dagger_\sigma$ ($\hat{d}_\sigma$) creates (annihilates) an electron of spin-$\sigma$ ($\sigma = \uparrow$ or $\downarrow$) on the impurity orbital of energy $\epsilon_d$; and $U$ is the intra-orbital electron–electron interaction energy. The reservoir hybridization functions assume a Lorentzian form of $\Lambda_\alpha(\omega) = \frac{\Delta_\alpha}{2} \frac{W_\alpha^2}{(\omega - \Omega_\alpha)^2 + W_\alpha^2}$, where $\Delta_\alpha$ is the effective coupling strength between the impurity orbital and the $\alpha$th reservoir ($\alpha = s$ or t), and $\Omega_\alpha$ ($W_\alpha$) is the band center (width) of the $\alpha$th reservoir.

The $K_{2'}$/Au(111) composite is also represented by a two-orbital AIM, in which the $d_\pi$ orbitals on each of the two Co adatoms are modeled by an impurity orbital, and the total impurity is described by

$$H_{imp} = \sum_{i=1,2} \left[ \epsilon_i (\hat{n}_{i\uparrow} + \hat{n}_{i\downarrow}) + U \hat{n}_{i\uparrow} \hat{n}_{i\downarrow} \right]. \tag{5}$$

The coupling between the $i$th orbital and $\alpha$th reservoir is characterized by the hybridization function $\Lambda_{ii\alpha}(\omega) = \frac{\Delta_{i\alpha}}{2} \frac{W_\alpha^2}{(\omega - \Omega_\alpha)^2 + W_\alpha^2}$. The CoPc-mold-mediated SE interaction between the two remotely separated local spins is quantified by the off-diagonal hybridization functions $\Lambda_{12s}(\omega) = \Lambda_{21s}(\omega) = \frac{\Delta_{SE}}{2} \frac{W_s^2}{(\omega - \Omega_s)^2 + W_s^2}$.

The values of the involving energetic parameters, including $\epsilon_d$, $U$, $\Delta_\alpha$, $\Omega_\alpha$, and $W_\alpha$, are extracted from the DFT calculation results (see Supplementary Table 2)[37,48,49].

**Hierarchical equations of motion method.** The HEOM method for fermionic baths[29,30] implemented in the HEOM–QUICK program[50] is employed to solve the stationary states of the AIMs, and to compute the key physical observables to compare directly with the experimental measurements. The detailed form of the HEOM can be found in ref. [29]. Its basic variables are the system reduced density matrix and a hierarchical set of auxiliary density operators (ADOs).

To ensure the numerical accuracy for ultralow temperatures, a recently developed Fano spectrum decomposition scheme[51] is adopted to accurately unravel the reservoir correlation functions. Limited by the computer resources at our disposal, the lowest temperature accessed by the HEOM method is 30 K for the single-orbital AIM, with the truncation tier set to $L = 4$. The tunneling current across a tip/$K_n$/Au(111) junction under a given bias voltage $V$ is extracted from the first-tier ADOs[29], and then the $dI/dV$ spectrum is obtained with a finite difference analysis.

We employ two approaches to theoretically determine the Kondo temperature $T_K$ of $K_n$/Au(111) composites. The first one makes use of an empirical scaling relation between the zero-bias conductance $G \equiv \left( \frac{dI}{dV} \right)_{V=0}$ of a Kondo impurity and the environmental temperature $T$ as follows[52,53]:

$$G(T) = G_0 \left[ 1 + \left( \frac{T}{T_K} \right)^2 \left( 2^{\frac{1}{s}} - 1 \right) \right]^{-s} + g_b. \tag{6}$$

Here, $G_0$ is the conductance at $T \to 0$, $g_b$ is the background conductance due to electron transport through the non-Kondo states, and $s$ is a parameter whose value

depends on the local spin state of the impurity. Particularly, $s = 0.22$ is adopted as suggested by previous calculations for a spin-$\frac{1}{2}$ impurity[52–54]. The Kondo temperature $T_K$ can thus be determined by fitting the calculated $G$ versus $T$ to Eq. (6).

The second approach is based on the relation of Eq. (2) from the Fermi liquid theory[55]. This requires the calculation of the full $dI/dV$ spectra for all the tip/$K_n$/Au (111) junctions. Since the environmental temperature $T$ adopted in the calculation is somewhat higher than the experimental counterpart, calculation needs to be done at a series of temperatures. The Kondo temperature $T_K$ is then extracted by fitting the resulting $\Gamma$ versus $T$ to Eq. (2) while taking $\lambda$ as a tunable parameter (see Supplementary Note 10 for more details).

## Data availability

The data that support the findings of this study are available from the authors on reasonable request; see author contributions for specific data sets.

## Code availability

The codes that were employed in this study are available from the authors on reasonable request.

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

## Acknowledgements

The support from the National Key Research and Development Program of China (Grant No. 2016YFA0200600), the National Natural Science Foundation of China (Grant Nos. 21973086, 21573202, 21633006, 21688102, and 21603205), the Strategic Priority Research Program of Chinese Academy of Sciences (Grant No. XDB36000000), and the Anhui Initiative in Quantum Information Technologies (Grant No. AHY090000) is greatly appreciated. The authors thank Xiaoli Wang, Yu Wang, Yao Wang, Xiangzhong Zeng, and Haoqi Chen for helpful discussions. The computational resources are provided by the Supercomputing Center of University of Science and Technology of China and Tianjin Supercomputer Center.

## Author contributions

J.Y., X.Z., and B.W. conceived the project. L.Z. and J.L. performed the experiments. X.L. conducted the theoretical calculations. X.L., X.Z., B.L., J.Y., B.W., Y.L., J.H., P.G., A.Z., and L.Y. analyzed the experimental and theoretical data. X.L., X.Z., J.Y., and B.W. co-wrote the paper. All authors discussed the results and commented on the paper.

## Competing interests

The authors declare no competing interests.
