## [Peer Review File · Nature Communications]

Reviewers' comments:

Reviewer #1 (Remarks to the Author):

In their paper the authors describe a detailed study on scanning tunneling spectroscopy (STS) studies of phthalocyanine (Pc) molecules adsorbed on a Au(111) substrate. The STS studies allows a clear demonstration of the presence of the Kondo effect. In going beyond previous work on essentially the same system (Ref. 45) by some of the authors of the present study, they not only investigate Pc-adsorbates containing only one Co atom but fabricate, using STM manipulation, also Pc adsorbates containing 2,3,4, and 5 Co atoms.

While the appearance in the STM of these differently metalated PC molecules is rather different, the resulting Kondo resonances are quite similar.

Although the data presented is of high quality and the theoretical analysis appears to be sound, in view of the rather large amount of papers on the Kondo effect of organometallic compounds adsorbed on metal substrates I cannot recommend the paper for publication in Nat. Comm. To me the Kondo spectra shown in Fig 2c look rather similar - it is not clear to me what advantage the Pc with 5 Co atoms has over a Pc with just 1 Co atom.

Reviewer #2 (Remarks to the Author):

In the manuscript "Molecular Molds for Regularizing Kondo States at Atom/Metal Interfaces" by YXiangyang Li and coworkers, the authors present an experimental and theory study of complexes between Co adatoms and CoPC molecules formed on an Au(111) surface. They use scanning tunneling microscopy to move the Co adatoms and CoPC molecules by tip induced manipulation over the substrate surface. By bringing a CoPC molecule close to individual Co adatoms, they observe the formation of a complex in which successively the lobes of the CoPC appears brighter by catching Co adatoms.

Using scanning tunneling spectroscopy they found on these complexes a sharp resonance close to zero bias when recording differential conductance versus bias traces. They interpret these resonances as a Kondo resonances originating from screening of the magnetic moment of the caught Co adatom by the many electrons of the Au(111) substrate. Interestingly, they observe a linear increase to the resonance line-width with the number of caught Co adatoms.

While I find the experimental results overall convincing and the quality of the manuscript sufficient for publication in Nature Communications, I have some minor comments on the experimental part but some serious questions concerning the theoretical model which I will detail below:

- Experimental Fano fit: Can the authors comment if they have included the temperature and modulation broadening into their fitting routine? In particular the high modulation voltage of 2mV (is this the effective modulation voltage?) will lead to a significant overestimation of the Kondo temperature.

- Figure 1: The black arrow and the "push" is almost invisible.

- Figure 2b: The distance dependent Kondo temperature seems to have an oscillatory behavior with distance. Is this significant? Please comment on this issue!

- Figure 3b: I found the determination of 2Δ very questionable. How is the well defined half-width at half-maximum (Δ) in the single-orbital AIM related to the spin-orbit and crystal-field induced spread of the five 3d-orbitals? Here the authors use some rather arbitrary cuts (zero signal?) to determine some 2Δ value. Note, that in an AIM the Kondo temperature depends exponentially on Δ . How is it possible that such simple model with such a crude and in my opinion not correctly assigned Δ lead to an extremely small error between experimental and theoretical values seen in

Figure 2d?

- Figure 3b (upper right inset): The explanation of the inset is insufficient. It is not clear why the energy is now scaled such that for "down" spins the dxz and dyz-orbitals are half filled, while this can not be taken out of the main panel of figure 3b. Furthermore, the inset suggest that the spin state of the Co is $S=1$ in contradiction to what the authors write in their main text, $S=1/2$. To clarify the spin state the authors could have used temperature dependent measurements in conjunction with equation (5). Did they performed such measurements? How sure are they that the spin is correctly assigned to $S=1/2$ or $S=1$?
- The authors claim that the increase of Kondo temperature in the complex K2' is due to a ferromagnetic coupling between the two impurities. What are the proves for this claim?
- Fig 4 inset show first a decay and then an increase of the Kondo temperature with superexchange energy. What does this mean? Is it that with low SE the Kondo-singlets get weakened by the FM coupling, however for higher SE a combined $S=1$ Kondo state appear? Please explain the physics behind this behavior.
- The authors write that the SE result into an effective FM coupling of $J_{\text{eff}} = 17\text{meV}$. This is in the same order as the Kondo scaling energy ($\sim k_{\text{BT}_K}$). Here complex physics should happen as for example the observation of triplet - single transitions. Why don't they observe them?

Reviewer #3 (Remarks to the Author):

The authors consider, both experimentally and theoretically, the effects of a molecular surrounding environment on the Kondo resonances associated to adsorbed Co atoms. A consistent stabilization of the Kondo resonance due to the molecules, as well as a systematic analysis of the Kondo temperature associated to different Co-molecule compounds is performed. The theoretical description moves from ab-initio calculations and studies the Kondo resonance within the framework of a Hierarchical equation of motion technique for the reduced density matrix. The manuscript is well written, presenting interesting results with a very solid analysis. I thus recommend its publication in its present form.

We thank the Reviewer for thoroughly reading our manuscript, and we appreciate his/her positive assessment on the quality of our experimental work and theoretical analysis.

The Reviewer's original comments are:

While the appearance in the STM of these differently metalated Pc molecules is rather different, the resulting Kondo resonances are quite similar.

Although the data presented is of high quality and the theoretical analysis appears to be sound, in view of the rather large amount of papers on the Kondo effect of organometallic compounds adsorbed on metal substrates, I cannot recommend the paper for publication in Nat. Comm. To me the Kondo spectra shown in Fig. 2c look rather similar – it is not clear to me what advantage the Pc with 5 Co atoms has over a Pc with just 1 Co atom.

The Reviewer considered our work to be somewhat similar to the existing papers which study the Kondo effect of organometallic compounds on metal substrates. We would like to point out that this is NOT the case – the present work aims to solve an important common problem existing in numerous previous works, i.e., the Kondo effect of adsorbed transition metal atoms varies sensitively to the surrounding environment. Moreover, despite the large amount of papers in the literature, it has remained largely unclear what leads to the observed sensitive dependence, because related theoretical studies were rather scarce and limited. Therefore, the achievement of our joint experimental and theoretical investigation is distinctly different from and much beyond the large amount of existing works. Detailed explanations are given in below.

(1) We agree with the Reviewer that there are already *a rather large amount of papers on the Kondo effect of organometallic compounds adsorbed on metal substrates*. However, it is important to note that many of the existing studies exhibited a common problem – the Kondo features vary greatly with the local chemical environment surrounding the transition metal atom. Such a problem makes it difficult to produce Kondo states at atom/metal interfaces with uniform features. We have illustrated this problem by mentioning two representative cases in the literature: a bare Co atom and an FePc molecule adsorbed at different locations on the Au(111) surface. Our work aims at making a complete change of the status quo by proposing a practical solution to the common problem. Such a goal clearly distinguishes our work from the large amount of papers in the literature.

To emphasize the uniqueness of our work, we have revised the introduction part of the manuscript with the following modified remarks:

“In many experiments, a TM adatom is bound to organic ligands to form an organometallic

compound, and its local electronic configuration is significantly influenced by the surrounding. . . . This poses a serious challenge for potential practical applications – is there a way to regularize the Kondo states, so that they could exhibit uniform features that are insensitive to the variation of environment?”

(2) We are glad that the Reviewer has noticed that *while the appearance in the STM of these differently metalated Pc molecules is rather different, the resulting Kondo resonances are quite similar*. Such a fact indicates the Kondo resonances are insensitive to the variation of environment beyond the Pc ring. This means that the goal of our work – producing Kondo states with uniform features – is achieved by the proposed molecular-mold strategy.

On the other hand, as has been pointed out in our manuscript, although *the Kondo spectra shown in Fig. 2(c) look rather similar*, the widths of the Kondo conductance peaks and hence the associated Kondo temperatures are distinctly different for the K_n complexes with different numbers of Co adatoms; see Fig. 2(d). Therefore, the real *advantage* of the constructed K_n complexes is that they enable the fine-tuning of Kondo effects by adjusting the number of Co adatoms and arranging their positions within a Pc ring.

To clarify the above point, and to accentuate the advantage of the proposed molecular-mold strategy, we have revised our manuscript to include the following remarks:

“These features indicate that the Kondo state centered at a captured Co adatom is affected subtly but regularly by the other Co adatoms confined within the same phthalocyanine ring. This lays the foundation for fine-tuning the Kondo states formed at atom/metal interfaces through precise arrangement of the TM adatoms.”

“To summarize, unlike all the previous studies in which the Kondo states formed around an adsorbed TM atom always vary sensitively to the surrounding environment, we demonstrate that, the proposed strategy of using a symmetric and conjugated molecular mold gives rise to highly robust and regularized Kondo states. Adjusting the chemical composition of the molecular mold further allows for a fine tuning of the Kondo characteristics. The resulting atom-mold complexes can move freely on the gold surface and exhibit remarkably uniform Kondo features. Therefore, they may serve as standard building blocks for the design and fabrication of novel quantum devices.”

(3) Last but not the least, we would like to mention another aspect in which our work has gone much beyond the previous works – it presents a comprehensive first-principles-based theoretical study on the constructed organometallic compounds. The calculation results, such as the dI/dV spectra and the theoretically determined Kondo temperatures, are compared side-by-side with the experimental values; see Fig. 2(d) of the revised manuscript and Fig. S10 of the revised Supplemental Information. This is in clear contrast to the large amount of papers in the literature, in which the Kondo states were either left out of calculation or characterized by rather simple models.

To highlight the above aspect, we have added the following sentence to the revised main text:

“... However, first-principles-based theoretical study on the Kondo states has remained rather scarce.”

To summarize, the above three aspects clearly distinguish our work from the large amount of papers in the literature. We have substantially revised our manuscript to highlight these important aspects. We thus hope the Reviewer could recognize the uniqueness and advantage of our proposed molecular-mold strategy, and deem our revised manuscript worthy of publication in Nature Communications.

We thank the Reviewer for thoroughly and carefully reading our manuscript, and for his/her overall positive assessment on the quality of our work. We are also grateful for the Reviewer's instructive comments and questions which have helped to improve substantially the presentation of our work.

The Reviewer's general critique is: *While I find the experimental results overall convincing and the quality of the manuscript sufficient for publication in Nature Communications, I have some minor comments on the experimental part but some serious questions concerning the theoretical model.*

Following the Reviewer's comments and questions, we have systematically re-examined our theoretical analysis. We have also carried out a large number of additional calculations. All these efforts have justified the validity of our theoretical model (to be elaborated in below). Accordingly, we have significantly revised our manuscript including the main text (with three updated figures) and the Supplemental Information (with four newly added figures).

Our point-to-point response to the Reviewer's comments and questions is as follows.

(1) *Experimental Fano fit: Can the authors comment if they have included the temperature and modulation broadening into their fitting routine? In particular the high modulation voltage of 2mV (is this the effective modulation voltage?) will lead to a significant overestimation of the Kondo temperature.*

We have explicitly included the temperature and modulation broadening into the experimental fitting routine. The sinusoidal modulation voltage used in our experiment has a peak-to-peak value of 2 mV, which corresponds to an amplitude of 1 mV. According to the literature, such as Sec. 3.1.1 of [*J. Phys.: Condens. Matter* **30**, 424001 (2018)], modulations with an amplitude below 1.4 mV have little impact on the measured resonance width. This is also confirmed by our analysis – because of the strong Kondo correlation of the $K_n/\text{Au}(111)$ composites, the thermal and modulation broadening gives rise to a small correction to T_K of only about 1.2 K.

To clarify this issue, we have added the following paragraph to the revised Methods section of the main text:

“The Kondo temperature T_K is extracted from Γ by deconvolution of the thermal and instrument-induced broadening as follows,

$$k_B T_K = \sqrt{\Gamma^2 - (\lambda k_B T)^2 - (0.87eV_m)^2}. \quad (1)$$

Here, k_B is the Boltzmann’s constant, $T = 5$ K is the environmental temperature, the coefficient λ assumes an empirical value of 2.7 as adopted in previous experiments, and $V_m = 1$ mV is the amplitude of the modulation voltage (corresponding to a peak-to-peak value of 2 mV).”

We have also added the following paragraph in Sec. ID of the revised Supplemental Information (SI) to address this issue:

“The influence of instrument on the measured Kondo temperature has been examined carefully in the literature. Since our experiments are conducted at a low temperature $T = 5$ K by using a small modulation voltage (peak-to-peak value is 2 mV and hence the amplitude is $V_m = 1$ mV), the thermal fluctuations ($k_B T$) and the modulation voltage (V_m) give rise to a rather minor contribution to the total broadening Γ . For instance, the measured broadening is $\Gamma = 10.6$ meV for the $K_1/\text{Au}(111)$ composite, we thus have $k_B T_K = \sqrt{\Gamma^2 - (1.16 \text{ meV})^2 - (0.87 \text{ meV})^2} = 10.5$ meV, i.e., the influence of thermal and instrument-induced broadening amounts to a small correction of T_K of only about 1.2 K. Even so, we have explicitly accounted for these minor contributions in Fig. 2 of the main text.”

(2) *Figure 1: The black arrow and the “push” is almost invisible.*

We have changed the color of arrow and the “push” to white; see the modified Fig. 1 of the main text.

(3) *Figure 2b: The distance dependent Kondo temperature seems to have an oscillatory behavior with distance. Is this significant? Please comment on this issue!*

We thank the Reviewer for drawing our attention to this interesting phenomenon. Indeed, the distance-dependent Kondo temperature has an oscillatory behavior with distance. Such an oscillatory behavior has been indicated in the revised Fig. 2(b) of the main text (see Fig. R1 in below).

Following the Reviewer’s suggestion, we have added the following paragraph to the revised main text to comment on this issue:

“From Fig. 2(a), an isolated K_1 has an almost constant T_K of about 122 K. By manipulating the STM tip, a K_1 can be moved towards a bare Co adatom. Throughout this process, the spectral line shape of the K_1 remains largely unchanged, while the T_K exhibits a weak oscillation around the constant value; see Fig. 2(b) and Fig. S3. Such an oscillation is supposed to originate from the substrate-mediated RKKY interaction between the Co adatom in the K_1 and the nearby bare Co adatom, and it is greatly suppressed with the nearby bare Co adatom captured by a CoPc mold.”

(4) *Figure 3b: I found the determination of 2Δ very questionable. How is the well defined half-width at half-maximum (Δ) in the single-orbital AIM related to the spin-orbit and crystal-field induced spread of the five 3d-orbitals? Here the authors use some rather arbitrary cuts (zero signal?) to determine some 2Δ value. Note, that in an AIM the Kondo*

FIG. R1. Variation of T_K of a K_1 (or a bare Co adatom) versus its distance to another K_1 (or bare Co adatom) on the Au(111) surface.

temperature depends exponentially on Δ . How is it possible that such simple model with such a crude and in my opinion not correctly assigned Δ lead to an extremely small error between experimental and theoretical values seen in Figure 2d?

[4a] The relation between the well defined half-width at half-maximum (Δ) in the single-orbital AIM and the environment-induced spread of $3d$ -orbitals has been discussed extensively in the literature, e.g. [*Phys. Rev. Lett.* **85**, 2557 (2000)]. To clarify such a relation for the K_n /Au(111) composites studied in this work, and to ascertain that the values of Δ are correctly assigned, we have added the following paragraph in the revised manuscript:

“The strength of Kondo correlation depends critically on how strongly the local spin on a Co adatom is screened by the electronic spins in the surrounding environment. For the K_n /Au(111) composites, the Kondo screening is realized via the hybridization of the Co d_π orbitals with the s orbitals of the nearby gold atoms. It is such s - d hybridization that leads to the broadening and splitting of the Co d_π orbitals as shown in Fig. 3(b) and Fig. S9. The hybridization strength is characterized by Δ_s , which is taken as the half width of the split PDOS peaks which constitute the main contribution to the local spin moment.”

Detailed explanations are provided in the newly added Sec. II D of the revised SI, as follows:

“The influence of spin-orbit coupling (SOC) on the electronic structures of K_n /Au(111) composites is examined by turning on non-collinear spin in the DFT calculation. It is found that aligning the local spin in different magnetization directions results in a rather small variation in the total energy (less than 0.3 meV per Co adatom), indicating that the SOC has a negligible effect on the local electronic structures of the K_n complexes. This is also consistent with the designation of a local $S = \frac{1}{2}$ state for each captured Co adatom.

In a $K_n/\text{Au}(111)$ composite, each Co adatom interacts with its surrounding environment, which consists of an isoindole ligand of the CoPc molecular mold and the gold substrate. In the following, we will elaborate on how the environment affects the local electronic structure of the Co adatom. In particular, we scrutinize the evolution of the PDOS of d_{xz} orbital, one of the d_π orbitals which contribute predominantly to the formation of Kondo states.

FIG. R2. (A) PDOS of the d_{xz} orbital of an isolated Co atom. (B) PDOS of the d_{xz} orbital of the captured Co atom in an isolated K_1 complex. The PDOS of the p_z orbitals of the carbon atoms on the adjacent isoindole ligand is also depicted. (C) PDOS of the d_{xz} orbital of the Co adatom in a $K_1/\text{Au}(111)$ composite. The PDOS of the s orbitals of the three nearest gold atoms in the $\text{Au}(111)$ substrate is also depicted.

For an isolated Co atom, the PDOS of its d_{xz} orbital exhibits a sharp peak; see Fig. R2(A). This affirms that the d_{xz} orbital is an eigen energy level of the Co atom. In an isolated K_1 complex, the Co adatom binds to an isoindole ligand of the CoPc mold through the d_π - π bonding interaction. The isoindole ligand field results in the rearrangement of the Co d orbitals and regularization of the local spin distribution (analogous to that depicted in Fig. 3(a) of the main text). On the other hand, it is evident from Fig. R2(B) that the PDOS of the d_{xz} orbital retains a single-peak structure – the main peak amounts to 90% of the total area. This suggests that the d_{xz} orbital of the Co atom largely preserves its atomic feature even under the influence of the ligand field.

Figure R2(C) plots the PDOS of the d_{xz} orbital of the Co adatom in a $K_1/\text{Au}(111)$ composite, whose line shape is distinctly different from those displayed in Fig. R2(A) and (B). The PDOS of the d_{xz} orbital is no longer a single peak. Instead, it has a rather scattered distribution over a wide energy range. Moreover, there is an apparent correspondence between the peak positions of the PDOS of Co d_{xz} orbital and those of s orbitals of the nearest gold atoms. Such a correspondence highlights a strong hybridization between the Co d_{xz} orbital and the Au s orbitals. It is this

hybridization that leads to the formation of Kondo states at the $K_1/\text{Au}(111)$ interface, since the latter is known to originate from the screening of local spin moment (contributed predominantly by the Co d_π orbitals) by the spins of itinerant electrons in the metal substrate (contributed by the Au s orbitals).

In general, the broadening (or splitting) of an atomic d orbital reflects its interaction with the surrounding environment. As shown in Fig. S6, the d_{xy} , $d_{x^2-y^2}$, and d_{z^2} orbitals of the Co adatom have much smaller broadening under the influence of the environment. Considering also the fact that these orbitals only present a minor contribution to the local spin moment (see Table S1), we conclude that it is the d_π (d_{xz} and d_{yz}) orbitals that contribute predominantly to the formation of Kondo states.

As shown in Fig. R2, the splitting of the d_{xz} orbital is caused solely by the hybridization with the s orbitals of surrounding gold atoms. Therefore, the energy span of the split d_π orbitals characterizes the s - d hybridization strength (Δ_s), which is a decisive factor of the strength of Kondo screening. Conventionally, the magnitude of Δ_s is extracted as the HWHM for the broadened or split d orbital. However, for the $K_n/\text{Au}(111)$ composites in our study, the PDOS of the split d_π orbitals has a complicated line shape, and thus we propose to extract Δ_s as half of the full width of the split peaks; see Fig. 3(b) of the main text.”

[4b] Regarding the determination of the values of Δ_s , we have added the following paragraphs in Sec. IIE of the revised SI to elaborate on the numerical procedure in detail:

“ Δ_s is the hybridization strength between the impurity and substrate orbitals, as it represents the amplitude of the hybridization function $\Lambda_s(\omega) \equiv \pi \sum_k |t_{kd}|^2 \delta(\omega - \epsilon_{ks}) = \frac{\Delta_s}{2} \frac{W_s^2}{(\omega - \Omega_s)^2 + W_s^2}$. In the wide-band limit ($\Delta_\alpha \ll W_\alpha$), the spectral function of the impurity orbital is

$$\rho_d(\omega) = -\frac{1}{\pi} \text{Im}[G_d^r(\omega)] \simeq \frac{1}{\pi} \frac{\Delta}{(\omega - \epsilon_d)^2 + \Delta^2}, \quad (2)$$

where $G_d^r(\omega)$ is the retarded Green’s function of the impurity orbital, and $\Delta = \Delta_s + \Delta_t \approx \Delta_s$.

Note that $\rho_d(\omega)$ is just the PDOS of the d_π orbitals of the captured Co adatoms, so Δ_s could be extracted as the HWHM of the broadened (or split) peak of $\rho_d(\omega)$, provided that $\rho_d(\omega)$ retains a single-peak structure. However, as shown in Fig. S9(C), $\rho_d(\omega)$ has a rather complicated line shape because of the strong s - d hybridization. Thus, Δ_s is extracted as the half width of the split peaks which constitute the major contribution to the local spin moment; see Fig. 3(b) in the main text. Numerically, the value of Δ_s is determined by cutting the split peaks with a sufficiently small threshold. For instance, the thresholds of 0.05 and 0.1 eV⁻¹ result in $\Delta_s = 0.261$ and 0.258 eV for the $K_1/\text{Au}(111)$ composite, respectively. It is estimated that the uncertainty of the cutting threshold could cause an uncertainty of ~ 5 K in the predicted Kondo temperature.”

[4c] We agree with the Reviewer that the error between experimental and theoretical values seen in Fig. 2(d) is remarkably small. Indeed, for an AIM the Kondo temperature depends

sensitively on Δ . Nevertheless, Δ appears not only in the exponent but also in the prefactor of the expression of T_K ; see [*Phys. Rev. Lett.* **109**, 266403 (2012)]. In particular, for AIMs away from the electron-hole symmetry point, the prefactor can have a rather complicated form. Therefore, the quantitative relation between Δ and T_K can only be determined by employing a highly accurate quantum impurity solver, such as the HEOM method used in this work. Moreover, as shown clearly in the insets of Fig. 3(b), the variation of Δ_s versus n apparently resembles the T_K versus n relationship, which also verifies our theoretical designation of the origin of Kondo resonances.

To assure the theoretically determined T_K are correct, we have carried out extra calculations on the dI/dV spectra of all the $K_n/\text{Au}(111)$ composites at different temperatures (see the new Fig. S12 of the revised SI), and the values of T_K are extracted from the widths of the Kondo resonance peaks. These new results have been added to the revised Fig. 2(d) of the main text (or see Fig. R3 in below).

FIG. R3. T_K of the K_n complexes extracted from experimentally measured and theoretically calculated dI/dV spectra. The dashed line is a linear fit of the measured T_K of K_n ($n = 1, 2, 3, 4$). The circles (triangles) represent the values of T_K determined by the heights (widths) of the calculated Kondo conductance peaks.

Accordingly, we have also added the following paragraphs in the Methods and Results sections of the revised main text to discuss these new theoretical results. More detailed discussions are provided in Sec. II F of the revised SI.

“We employ two approaches to theoretically determine the Kondo temperature T_K of $K_n/\text{Au}(111)$ composites. The first one makes use of an empirical scaling relation between the zero-bias conductance $G \equiv (\frac{dI}{dV})_{V=0}$ of a Kondo impurity and the environmental temperature T as follows. \dots The second approach is based on the relation of Eq. (1) from the Fermi liquid theory. This requires the calculation of the full dI/dV spectra for all the tip/ $K_n/\text{Au}(111)$ junctions. Since the environmental

temperature T adopted in the calculation is somewhat higher than the experimental counterpart, calculation needs to be done at a series of temperatures. The Kondo temperature T_K is then extracted by fitting the resulting Γ versus T to Eq. (1) while taking λ as a tunable parameter; see the SI for more details.”

“The calculated dI/dV spectra agree remarkably with the experimental measurements (see Fig. S10 in the SI). The T_K of K_n are determined based on the height or the width of the Kondo conductance peaks (see Methods and also Figs. S11 and S12 in the SI), and the theoretical values accurately and consistently reproduce the experimental data; see Fig. 2(d).”

(5) *Figure 3b (upper right inset): The explanation of the inset is insufficient. It is not clear why the energy is now scaled such that for “down” spins the d_{xz} and d_{yz} -orbitals are half filled, while this can not be taken out of the main panel of figure 3b. Furthermore, the inset suggest that the spin state of the Co is $S = 1$ in contradiction to what the authors write in their main text, $S = \frac{1}{2}$. To clarify the spin state the authors could have used temperature dependent measurements in conjunction with equation (5). Did they performed such measurements? How sure are they that the spin is correctly assigned to $S = \frac{1}{2}$ or $S = 1$?*

To clarify the issue raised by the Reviewer and to avoid any possible confusion, we have modified Fig. 3(b) and simplified the upper right inset (see Fig. R4 in below).

From the upper right inset of Fig. R4, it is clear that d_{xz} and d_{yz} orbitals are not half filled because of the partial occupation of spin-down electrons (represented by the dotted arrows). Instead, the two d_π orbitals together carry a spin-unpaired electron, which amounts to a local spin moment of $S_z = \frac{1}{2}$; see Table S1 in the SI. Moreover, the designation of a local $S = \frac{1}{2}$ state for each Co adatom is fully consistent with the results of HEOM calculation; see also our response to the Reviewer’s Point (6) in below.

To clarify this issue, we have added the following paragraphs to Sec. II B of the revised SI:

“Table S1 lists the electron occupation numbers of the d orbitals on the captured Co adatom in a K_1 /Au(111) composite, along with their contributions to the local spin moment S_z . In distinct contrast to the Co ion in the CoPc mold which has a negligible S_z , the captured Co adatom has an appreciable value of $S_z \approx 0.5$, which originates mainly from the d_π (d_{xz} and d_{yz}) orbitals. Thus, the Co adatom is considered to be in a local $S = \frac{1}{2}$ state.

Note that a halfly occupied orbital often refers to the scenario that the orbital is occupied by a spin-up electron with no spin-down electron occupancy, and so the overall spin moment is $S_z = \frac{1}{2}$. In contrast, for the captured Co adatom in a K_1 , each of the d_π orbitals is occupied by roughly one spin-up electron but also by half a spin-down electron; see Table S1 as well as the inset of Fig. 3(b) in the main text. Because of the fractional (nonzero) occupation of spin-down electrons, each d_π orbital contributes only about $s_z = 0.21$ to the local spin moment, and it is the sum of s_z of all the five Co d orbitals that amounts to the spin moment ($S_z = \frac{1}{2}$) of a purely halfly occupied orbital.”

FIG. R4. PDOS of d_π orbitals of the captured Co adatom in a K_1 . Positive (negative) values represent spin-up (spin-down) electrons. Δ_s is the half width of the split peaks which constitute the main contribution to the local spin moment. The upper right inset is an orbital diagram showing the relative position and electron occupancy of every d orbital of the Co adatom, where the thickness of a line indicates the broadening of the orbital by its surrounding environment, and a solid (dotted) arrow represents one (half an) electron residing on the orbital. The lower left inset displays the PDOS of d_π orbitals of a captured Co adatom in each of the K_n complexes ($n = 1, 2, 2', 3, 4$). The dashed lines mark the full width of the main peaks, which are shifted and aligned horizontally for a clear comparison. The lower right inset depicts Δ_s versus n .

We have not performed experimental measurement to determine the local spin state because there have been studies in the literature showing that this is very difficult. For instance, it was found in [Nat. Commun. 2, 490 (2011)] that the dI/dV spectra of a CuPc/Ag(100) composite can be fit to both the $S = \frac{1}{2}$ and the $S = 1$ Kondo models. Therefore, in our work the local spin state of the Co adatoms is determined only by theoretical means.

(6) *The authors claim that the increase of Kondo temperature in the complex $K_{2'}$ is due to a ferromagnetic coupling between the two impurities. What are the proves for this claim?*

The existence of a ferromagnetic coupling between the two Co adatoms in the complex $K_{2'}$ is supported by the calculation results obtained by both the DFT and HEOM methods. Regarding the numerical evidence from the DFT calculations, we have added the following paragraphs in Sec. II B of the revised SI, along with the newly added Fig. S7 (see Fig. R5 in below):

“For the $K_{2'}$ /Au(111) composite in which two Co adatoms locate at the para positions of the Pc ligand, the local spins on the two distantly separated Co adatoms can be parallel or anti-parallel to each other, corresponding to a ferromagnetic (FM) or antiferromagnetic (AFM) spin-state of the composite, respectively.

Because of the long-range SE interaction between the two local spins mediated by the delocalized Kohn-Sham orbitals on the CoPc mold (see Fig. S4), there is a finite energy gap between the FM and AFM spin-states, $J_{\text{eff}} = E_{\text{AFM}} - E_{\text{FM}} \approx \gamma \frac{\Delta_{\text{SE}}^2}{U + \Delta\epsilon} = 14 \text{ meV}$, where $\Delta\epsilon = |\epsilon_d - \epsilon_{\text{channel}}| = 0.205 \text{ eV}$, $\Delta_{\text{SE}} = 0.123 \text{ eV}$, $U = 0.88 \text{ eV}$ (see Sec. II E), and $\gamma = 1$. It is worth mentioning that such a value of J_{eff} is but a crude estimate, because of the potential uncertainty in the prefactor γ . DFT calculation using the PBE functional yields an energy gap of $J_{\text{eff}} = 4 \text{ meV}$, and the spin density distributions corresponding to both the FM and AFM spin-states of the $\text{K}_{2'}/\text{Au}(111)$ composite are shown in Fig. R5.”

FIG. R5. Spin density distribution of a $\text{K}_{2'}/\text{Au}(111)$ composite in the ferromagnetic (left panel) and antiferromagnetic (right panel) states. The value of isosurface is $\pm 0.05 \text{ \AA}^{-3}$.

Regarding the numerical evidence from the HEOM calculations, we have added the following paragraphs in Sec. III D of the revised SI, along with the newly added Fig. S18 (see Fig. R6 in below):

FIG. R6. Calculated $S_{12} \equiv \langle \hat{S}_1 \cdot \hat{S}_2 \rangle$ and $\langle \hat{S}^2 \rangle$ versus Δ_{SE} for a $\text{K}_{2'}/\text{Au}(111)$ composite. A two-orbital AIM which explicitly involves Δ_{SE} is adopted.

“As described in Sec. II B of SI, each of the captured Co adatoms in a $\text{K}_n/\text{Au}(111)$ composite is in a local $S = \frac{1}{2}$ state. Since it is the d_π orbitals that contribute predominantly to the Kondo screening, the local spin associated with the Kondo states is $S_z \simeq 0.42$.

For describing the Kondo states in the $\text{K}_{2'}/\text{Au}(111)$ composite, we adopt a two-orbital AIM which involves an off-diagonal hybridization function, $\Lambda_{12}(\omega) = \frac{\Delta_{\text{SE}}}{2} \frac{W_s^2}{(\omega - \Omega_s)^2 + W_s^2}$, to characterize the long-range SE interaction between the two Co adatoms mediated by the CoPc mold. To verify that the SE interaction indeed results in an effective FM coupling between the two Co adatoms, we

compute $S_{12} = \langle \hat{\mathbf{S}}_1 \cdot \hat{\mathbf{S}}_2 \rangle$ and $\langle \hat{\mathbf{S}}^2 \rangle$ as functions of Δ_{SE} by using the HEOM method. Here, $\hat{\mathbf{S}}_i = \frac{1}{2} \sum_{ss'} \hat{c}_{is}^\dagger \boldsymbol{\sigma}_{ss'} \hat{c}_{is'}$ is the spin operator for the d_π orbitals of the i th Co adatom ($i = 1, 2$), with $\boldsymbol{\sigma}$ representing the set of Pauli matrices; and $\hat{\mathbf{S}} = \hat{\mathbf{S}}_1 + \hat{\mathbf{S}}_2$. The calculated results are shown in Fig. R6.

As shown in Fig. R6, the spin-spin correlation S_{12} increases monotonically from zero to a positive value with the increasing Δ_{SE} . This indicates that the two remotely separated local spins are independent of each other at $\Delta_{\text{SE}} = 0$, while they are aligned toward a mutually parallel direction at a nonzero Δ_{SE} . It is noted that two lines almost overlap each other in Fig. R6. This is because the local spin moments on the Co adatoms (S_1 and S_2) remain almost constant upon the variation of Δ_{SE} . By using the relation $\langle \hat{\mathbf{S}}^2 \rangle = S(S + 1)$, we have $\langle \hat{\mathbf{S}}_2^2 \rangle \approx \langle \hat{\mathbf{S}}_1^2 \rangle = S_1(S_1 + 1) = \sqrt{(\langle \hat{\mathbf{S}}^2 \rangle - 2S_{12})/2}$, and thus $S_1 \approx S_2 = 0.4$. Therefore, the results yielded by the HEOM calculation on the two-orbital AIM agree closely with those of the DFT calculation, cf. Table S1.”

(7) *Fig 4 inset show first a decay and then an increase of the Kondo temperature with superexchange energy. What does this mean? Is it that with low SE the Kondo-singlets get weakened by the FM coupling, however for higher SE a combined $S = 1$ Kondo state appear? Please explain the physics behind this behavior.*

Following the Reviewer’s suggestion, we have added the following paragraph in the revised main text to explain the physics behind this behavior:

“As shown in Fig. 4, the T_K of a $\text{K}_{2'}(\text{CoPc})$ exhibits a nonmonotonic dependence on Δ_{SE} . Such a phenomenon can be understood as follows. As Δ_{SE} increases from zero, the local spin on a Co adatom starts to feel the ferromagnetic interaction from the other Co adatom, and thus it is screened less strongly by the surrounding environment, which leads to the attenuated Kondo correlation. However, when Δ_{SE} reaches a certain value (see Fig. S18 in the SI), the whole $\text{K}_{2'}(\text{CoPc})$ complex favors an $S = 1$ state, resulting in a substantially enlarged spin moment for the total complex. Consequently, the Kondo temperature rises again with further increasing of Δ_{SE} .”

We have also provided more discussions in Sec. III D of the revised SI, as follows:

“In a serially coupled two-orbital AIM, one impurity orbital is coupled to only one electron reservoir. The whole impurity is described by $H_{\text{imp}} = \sum_{i=1,2} [\epsilon_i (\hat{n}_{i\uparrow} + \hat{n}_{i\downarrow}) + U \hat{n}_{i\uparrow} \hat{n}_{i\downarrow}] + t \sum_{\sigma=\uparrow,\downarrow} (\hat{d}_{1\sigma}^\dagger \hat{d}_{2\sigma} + \text{H.c.})$. Here, the last term represents the inter-orbital hopping (with t being the strength), which gives rise to an effective AFM coupling between the two orbitals, with the coupling strength being $J_t = \frac{4t^2}{U}$. Numerical calculations have revealed that the Kondo effect exhibits a non-monotonic trend with the increase of t . Specifically, the Kondo effect is first enhanced as t increases from zero to a certain value, and then it gets suppressed as t increases further. The mechanism of such a non-monotonic behavior is as follows. When t increases from zero, the local spin on one impurity orbital starts to feel the AFM interaction from the other impurity orbital, and thus it is screened more strongly by its surrounding environment. This results in enhanced Kondo correlation. On the other hand,

when t reaches a certain value, the two local orbitals start to form delocalized molecular orbitals. Because of the AFM nature of the inter-orbital coupling, this results in an $S = 0$ ground state within the impurity, leading to the quenching of local spins. Consequently, the Kondo correlation is more and more suppressed with further increasing of t .

The situation is exactly opposite for the $K_{2'}/\text{Au}(111)$ composite. Instead of the AFM t -coupling, the environment-induced SE interaction is of FM nature. Thus, when Δ_{SE} increases from zero, the local spin on one impurity orbital starts to feel the FM interaction from the other impurity orbital, and thus it is screened less strongly by its surrounding environment. This results in weakened Kondo correlation. Nevertheless, when Δ_{SE} reaches a certain value, the whole impurity favors an $S = 1$ ground state because of the FM coupling, resulting in a substantially enlarged spin moment within the total impurity. Consequently, the Kondo correlation is strengthened with further increasing of Δ_{SE} . This thus explains the non-monotonic variation of T_{K} versus Δ_{SE} depicted in Fig. 4 of the main text.”

(8) *The authors write that the SE result into an effective FM coupling of $J_{\text{eff}} = 17 \text{ meV}$. This is in the same order as the Kondo scaling energy ($\sim k_{\text{B}}T_{\text{K}}$). Here complex physics should happen as for example the observation of triplet-single transitions. Why don't they observe them?*

We thank the Reviewer for raising this insightful question, which has urged us to re-examine the evaluation of J_{eff} . We would like to point out this is indeed a highly challenging problem for the present quantum chemistry methods. Our original estimation of $J_{\text{eff}} \approx \frac{\Delta_{\text{SE}}^2}{U} = 17 \text{ meV}$ was a bit too crude, because the analytic expression is a simplified form and is subject to large uncertainties in the prefactor and denominator. A modified estimation that accounts for the energy difference between the Co d_{π} orbitals and the delocalized Kohn-Sham orbital is $J_{\text{eff}} = E_{\text{AFM}} - E_{\text{FM}} \approx \gamma \frac{\Delta_{\text{SE}}^2}{U + \Delta\epsilon} = 14 \text{ meV}$, but the prefactor γ is still subjected to a large uncertainty. We have further calculated J_{eff} by using both the DFT and HEOM methods, and they consistently yield values of $J_{\text{eff}} \approx 4 \text{ meV}$. Such a J_{eff} is much smaller than the Kondo energy scale $k_{\text{B}}T_{\text{K}} = 16.5 \text{ meV}$. This thus explains why the triplet-single transitions cannot be observed in the dI/dV spectra.

To address this issue, we have added the following paragraphs in Sec. II B and Sec. III D of the revised SI:

“DFT calculation using the PBE functional yields an energy gap of $J_{\text{eff}} = 4 \text{ meV}$, and the spin density distributions corresponding to both the FM and AFM spin-states of the $K_{2'}/\text{Au}(111)$ composite are shown in Fig. R5. Since the GGA functional may suffer from static correlation error, calculation of J_{eff} by more accurate *ab initio* quantum chemistry methods is appealing. This is however rather challenging because of the large size of the $K_{2'}/\text{Au}(111)$ composite.”

“As mentioned in Sec. II B, the long-range SE interaction leads to a finite energy gap J_{eff} between

the FM and AFM states of the $K_2'/\text{Au}(111)$ composite. A crude estimation gives $J_{\text{eff}} = E_{\text{AFM}} - E_{\text{FM}} \approx \gamma \frac{\Delta_{\text{SF}}^2}{U + \Delta\epsilon} = 14 \text{ meV}$, but the prefactor γ does not have a simple form and thus its value is subject to a large uncertainty.

To have a more accurate assessment of J_{eff} , we explicitly include the effective AFM t -coupling into the two-orbital AIM, and finite-tune the value of t until the resulting $S_{12} = 0$. The latter indicates that the FM and AFM spin-states become degenerate. From the HEOM calculation the zero spin correlation is reached at $t \simeq 0.027 \text{ eV}$. We thus have $J_{\text{eff}} - J_t = E_{\text{AFM}} - E_{\text{FM}} = 0$, and so $J_{\text{eff}} = J_t = \frac{4t^2}{U} \simeq 3.3 \text{ meV}$. Such a value agrees consistently with the value of 4 meV calculated by using the DFT method. Since $J_{\text{eff}} < k_{\text{B}}T_{\text{K}}$ (the latter is about 16.5 meV), the triplet-to-singlet spin excitation is overwhelmed by the Kondo resonance and is thus invisible in the dI/dV spectra.”

Besides the above modifications recommended by the Reviewer, we have also made various changes throughout the manuscript to further enhance its readability. With the substantial modifications made as above, the theoretical analysis presented in our work should be much more convincing, and thus the revised paper would be easily understood by the general readers of Nature Communications.

REVIEWERS' COMMENTS:

Reviewer #1 (Remarks to the Author):

I have carefully studied the reply of the referees to my original comment as well as to those made by reviewer # 2. The authors have considered all the points raised in my initial report. After reading through them, I did not change my original opinion. The effects observed in the present paper – in view of the existing rather large amount of papers on Kondo effect– are really very subtle - the data themselves do not show an obvious effect (as admitted by the authors). One really has to go to a more detailed analysis of the data, e.g. by determining the half width from a fitting procedure, see figure 2c.

I have read also the reply of the authors to the comments made by referee 2, which were rather numerous. From studying these concerns together with the replies made by the authors I feel that although the data are interesting in principle the analysis of the data is somewhat ambiguous. I think the paper in its present form should be published in a more specialized journal, e.g. Phys. Rev. B. I do not see the substantial advancement in physical or chemical insight required for a publication in Nature Communications.

Reviewer #2 (Remarks to the Author):

In the revised version the authors have extended and clarified the manuscript significantly and therefore I can now recommend publishing the paper.

I also believe, contrary to referee (1), that it contains enough novelty to be published in the Nature Communication journal.

However, I would like to recommend the authors to change the "equal" in equation (2) to an "approximation", i.e. " \approx ".

In general the temperature and modulation broadening of the Kondo feature (or any other spectroscopic feature) should be correctly done by a convolution of the expected signal with the different broadening kernels. For finite temperature the kernel is $[(1-f(x))*f(x)]$ with $f(x)$ the appropriate Fermi-Dirac distribution, while the bias modulation has a half-spherical broadening kernel of $\text{real}(\sqrt{1-x^2})$ with $x=V_{\text{bias}}/V_{\text{mod}}$.

I'm aware that many people are using the approximated equation (2), but from a didactic viewpoint it should nevertheless clearly marked as an approximation.

Markus Ternes

We thank the Reviewer for recommendation of publishing our paper in the Nature Communications, and for his recognition of the novelty of our work. We are also grateful for the Reviewer's suggestion to modify Eq. (2) in the manuscript. Our detailed response to the Reviewer's instructive comment is as follows.

(1) *However, I would like to recommend the authors to change the “equal” in equation (2) to an “approximation”, i.e. “ \approx ”. In general the temperature and modulation broadening of the Kondo feature (or any other spectroscopic feature) should be correctly done by a convolution of the expected signal with the different broadening kernels. For finite temperature the kernel is $[(1 - f(x)) * f(x)]$ with $f(x)$ the appropriate Fermi-Dirac distribution, while the bias modulation has a half-spherical broadening kernel of $\text{real}(\text{sqrt}(1 - x^2))$ with $x = V_{\text{bias}}/V_{\text{mod}}$. I'm aware that many people are using the approximated equation (2), but from a didactic viewpoint it should nevertheless clearly marked as an approximation.*

Following the Reviewer's suggestion, we have changed “=” to “ \approx ” in Eq. (2). In addition, we have added the following paragraph in the Methods section of the revised main text to further clarify this issue.

“It is worth pointing out that in principle the thermal and instrument-induced broadening of any spectroscopic feature should be examined by a convolution of the expected signal with a proper broadening kernel. In practice it is often more convenient to use the approximate formula of Eq. (2).”